# What Keeps Them Physically Active? Predicting Physical Activity, Motor Competence, Health-Related Fitness, and Perceived Competence in Irish Adolescents after the Transition from Primary to Second-Level School

**DOI:** 10.3390/ijerph17082874

**Published:** 2020-04-21

**Authors:** Una Britton, Johann Issartel, Jennifer Symonds, Sarahjane Belton

**Affiliations:** 1School of Health & Human Performance, Faculty of Science & Health, Dublin City University, D09 V209 Dublin 9, Ireland; johann.issartel@dcu.ie (J.I.); sarahjane.belton@dcu.ie (S.B.); 2School of Education, Social Sciences & Law, University College Dublin, D04 V1W8 Dublin 4, Ireland; jennifer.symonds@ucd.ie

**Keywords:** physical activity, motor competence, health-related fitness, perceived competence, school transition, youth

## Abstract

Physical activity (PA) decreases with age. The school transition is noted for significant changes in PA behaviour. Motor competence (MC), health-related fitness (HRF), and perceived competence (PC) are generally positively associated with PA. The aim of this study was to examine longitudinal cross-lagged relationships between PA, MC, HRF, and PC across the school transition from final year of primary school to first year of second-level school in Irish youth. PA (accelerometery), object-control and locomotor MC (TGMD-III), PC (perceived athletic competence subscale of the Self-Perception Profile for Adolescents), and HRF (20 m shuttle run, horizontal jump, vertical jump, push-ups, curl-ups) were measured in final year of primary school (6th class) and first year of second-level school (1st year). In the sixth class, 261 participants (53% female; mean age 12.22 ± 0.48 years) were tested. In first year, 291 participants (48% female; mean age: 13.20 ± 0.39 years) were tested. In total, 220 participants were involved in the study at both timepoints. Cross-lagged regression in AMOS23, using full information maximum likelihood estimation, was conducted to test reciprocal and predictive pathways between variables. The full cross-lagged model showed acceptable fit (χ^2^ = 69.12, df = 8, *p* < 0.01, NFI = 0.93, CFI = 0.94). HRF was the strongest predictor of future PA (β = 0.353), and also predicted PC (β = 0.336) and MC (β = 0.163). Object-control MC predicted future PA (β = 0.192). Reciprocal relationships existed between object-control MC and PA, and between object-control MC and PC. HRF was the strongest predictor of PA. Object-control MC also predicted PA. PA promotion strategies should target the development of HRF and object-control MC in primary school to reduce the decline in PA frequently observed after the school transition.

## 1. Introduction

Physical activity (PA) is associated with positive health outcomes in youth [1,2], but globally there is a trend for decreasing PA with age [1,3,4]. Despite recognition of the importance of PA, and subsequent public health initiatives targeting the problem of youth physical inactivity, there is little to suggest this decline in PA is being reversed. A key period where substantial changes in PA are often reported is the transition from primary to second-level school [4,5]. In Irish youth, students in second-level school have been found to be significantly less active than their primary school counterparts [6,7]. In addition, a significant sex difference is frequently reported, with males more active than females [8,9]. The school transition in Ireland, as in several other countries, occurs at the age of approximately 12 years, often coinciding with the onset of puberty. Notwithstanding the significant physiological changes that occur at this time, changing school environment, as an independent factor, has been shown to impact on PA behaviours [5,10,11].

Numerous factors are associated with youth engagement in PA [12,13]. A seminal model published by Stodden and colleagues in 2008 [13], which includes motor competence (MC), health-related fitness (HRF), and perceived competence (PC) as factors associated with PA, has since guided a body of research within the PA domain seeking to understand pathways that promote engagement in PA. Within this model MC is positioned as a primary determinant of PA, with HRF and PC proposed to mediate the relationship between MC and PA [13]. Pathways between each of the variables are hypothesised to be reciprocal and developmental, in that each variable has an impact on each other variable and the strength of these associations is likely to change with age [12,13].

MC is generally used to describe goal-directed movement requiring coordination of the body [14]. An important aspect of MC is proficiency in fundamental movement skills (FMS) [13]. FMS are considered the basic building blocks needed for future PA [15] and include locomotor skills, such as running, skipping and jumping, and object-control skills such as kicking, throwing and striking [16]. Longitudinally, MC has been shown to be positively associated with PA, with low-MC children often becoming low-active adolescents [17,18,19]. It is generally accepted that children have the capacity to master the basic components of most FMS by the age of between 6 and 9 years [16]. Recent research however shows that many young people are not reaching the expected level of FMS mastery, both nationally [20,21] and internationally [22]. Sex differences in MC are also frequently reported, with females consistently found to be less competent in motor skills than males [8,17,23]. Considering the separate components that make up MC, females consistently score lower in object-control MC compared to males [23,24]. Sex differences have not been found to be as pronounced in locomotor MC with some studies finding no sex difference in locomotor MC [23,24], while others report females as more proficient [25].

HRF, a proposed mediator in Stodden et al.’s (2008) [13] model, generally refers to a multidimensional construct consisting of cardiorespiratory endurance (CRE), muscular strength (MS), muscular endurance (ME), and flexibility [26]. Across all components of HRF, males have consistently been shown to outperform females [27,28]. Often, researchers choose just one component of HRF as a measure of fitness, with CRE most frequently assessed [7,29,30] due to the well-established positive association between CRE and health [31]. There is a growing body of evidence that supports a positive association for MS and ME with health [32], indicating that MS and ME, in addition to CRE should be measured in youth. Longitudinally, childhood HRF is found to be a positive predictor of adolescent PA [33,34], while cross-sectionally, HRF is also positively associated with MC in youth [14]. Current trends for HRF report that the majority of youth (>60%) are achieving acceptable levels of CRE [9,28]. That being said, Tomkinson et al. (2018) [35] found that for every year from the age of nine, the percentage of males and females achieving a healthy CRE level decreased by 3% and 7%, respectively. Considering the strong evidence for a positive association between HRF and health in youth, and cognisant of the drop-off in PA frequently reported during adolescence, establishing the role of HRF at this stage of development may be an important step in understanding how and why youth disengage from PA.

PC refers to an individual’s ability to effectively master a task [36] and, like HRF, is positioned as a mediator of the MC-PA relationship within Stodden et al.’s (2008) [13] conceptual model. Cross-sectionally, positive associations between PC and self-report PA in youth have frequently been reported [37,38]. When assessing PA using objective measures however, the same positive association has not always been found [39,40]. In relation to MC, a positive association between PC and MC has been identified in youth, with some longitudinal studies finding that MC positively predicts PC in older adolescents [41]. Sex and age have been reported to influence PC levels in youth. Males are frequently found to have higher PC levels than females [8,42], and PC has been found to be higher in younger compared to older youth [8]. In some cases, PC has been found to be negatively affected by school transition [43]. However, a review of the effects of school transition on psychological development generally highlighted how there are numerous interacting factors which determine whether the school transition has a positive or negative impact on individuals [44]. Research on the development of PC in the physical domain within an Irish setting is scarce, particularly in relation to changes during the school transition. Considering the significant impact that the school transition may have on PC [43] and the role that PC may play in promoting future PA [13], it is important to establish the nature of the relationship between PC and PA over the course of the school transition, with a view to reducing the negative impact of this transition on future PA.

Individual pathways outlined in Stodden et al.’s (2008) [13] conceptual model have been tested, both cross-sectionally and longitudinally. However, the complexity of the proposed model, particularly the hypothesised reciprocal pathways, requires a longitudinal design including all the variables identified by Stodden et al. (2008) [13] to fully understand the predictive and reciprocal nature of the pathways. In addition, despite the frequently reported differences between males and females in PA, MC, HRF and PC, the model does not account for sex differences in its hypothesised pathways. The current study aims to evaluate the pathways outlined in Stodden et al.’s (2008) [13] conceptual model, taking account of potential differences by sex, during a critical period in a child’s life, the transition from primary to second-level school. Using cross-lagged panel analysis this study aims to gain an understanding of the primary predictors of PA post school transition. The main research hypotheses are; (1) MC and HRF in the sixth class will predict MVPA in first year, (2) sex differences will exist in the nature of the predictive pathways between variables, (3) relationships between variables across the school transition will be reciprocal.

## 2. Materials and Methods

### 2.1. Participants

Data were collected from participants in the 6th class of primary school (January–April 2017), and one year later in 1st year of second-level school (January–May 2018). Participant recruitment began with contacting second-level schools which fit the criteria of having a maximum of three main feeder primary schools. Two second-level school principals consented for their schools to be involved in the study. Following this, the main feeder primary schools for both consenting second-level schools were contacted. Six primary school principals (three feeder schools for each second-level school) provided consent. Parental consent and participant assent were obtained in the 6th class of primary school for each participating student. At baseline, 261 participants were tested (53% female and 47% males; mean age 12.22 ± 0.48 years). Of these, 41 participants were not available for testing in 1st year of second-level school as they did not transfer into the catchment area schools selected for data collection. In 1st year, 291 participants (48% female and 52% male; mean age: 13.20 ± 0.39 years) were tested. The sample in 1st year included an additional 71 students who had not attended the feeder primary schools. These additional participants provided assent, and parental consent, prior to data collection in 1st year. In total, 335 participants were involved in the study for at least one timepoint, and 220 of these participated at both timepoints. Ethical approval for all measures was obtained from the authors’ institutional ethics committee (ethical approval number: DCUREC2016_109).

### 2.2. Measures

Actigraph (Actigraph LLC, Pensacola, FL, USA) accelerometers (models: GT1M, GT3X, GT3X+, wGT3X-BT), set to capture data in 10-s epochs, were used to measure PA [45]. Participants wore the accelerometer for nine consecutive days during waking hours. To account for subject reactivity the first and last days of the wear period were excluded from analysis [46]. A reminder text message was sent each morning to participants who had provided a contact phone number to increase compliance [47]. Minutes of moderate-vigorous PA (MVPA) per day were calculated using Evenson et al. (2008) cut-points [48] using Actilife (Actigraph LLC, Pensacola, FL, USA) version 6.13.3. Non-wear time was identified as greater than 20 min of zero-counts, and count values of <0 and ≥15,000 per minute were excluded [45]. Similar to other studies measuring PA across school transition, greater than or equal to 8 h was considered a valid day [5,49]. The minimum number of valid days required for inclusion in analysis were two week- and one weekend day [5].

MC was assessed using skills selected from the 3rd edition of the Test of Gross Motor Development(TGMD-III [50] and the Victoria Department of Education (1996) manual [51]: (1) object control skills: kick, catch, overhand throw, one-hand strike, and two-hand strike, (2) locomotor skills: run, skip, horizontal jump, and vertical jump. Each skill is made up of movement components. The presence or absence of a component is scored with a 1 or 0 respectively. Participants were videoed performing each skill and these videos were later scored by trained researchers.

Harter’s Self-Perception Profile for Adolescents (SPPA) [52] was used to measure perceived athletic competence (PAC). Athletic competence is one of nine dimensions in the SPPA, with the others being scholastic competence, social competence, physical appearance, job competence, romantic appeal, behavioural conduct, close friendship, and global self-worth. Each dimension is measured using five items scored with a structured alternative format scale designed to reduce social desirability [52]. Each item has two criteria, e.g., “*Some kids feel that they are better than others their age at sports BUT other kids don’t feel they can play as well*”. Participants first select which criteria they fit best with, then score this as either “*sort of true for me*” or “*really true for me*”, creating a four-point scale. In the PAC dimension, items regard how well participants feel they do at sport, and how athletic they are in general. Higher scores indicate more positive self-perceptions [52]. At both timepoints the PAC subscale showed satisfactory internal reliability (Cronbach’s alpha coefficient; 0.83 and 0.87), composite reliability (CR; 0.85 and 0.88), and convergent validity (Mean Variance Extracted: 0.49 and 0.56) in 6th class and 1st year respectively. Values greater than or equal to 0.70 for Cronbach’s alpha and CR indicate acceptable reliability, while values greater than or equal to 0.50 for mean variance extracted indicate acceptable convergent validity [53].

HRF was measured using five tests (20 metre shuttle run, horizontal jump, vertical jump, push-ups and curl-ups). A previous confirmatory factor analysis (CFA) showed these five tests give a fair representation of HRF in youth [54]. Protocols for each HRF test can be found in manuals for FITNESSGRAM [55] and EUROFIT [56], and in the HELENA study [27,57] (Table 1).

### 2.3. Data Processing

Due to the longitudinal study design, a missing data analysis was conducted. All variables had some degree of missingness at both timepoints. In 6th class 220 participants (65%) had missing data on at least one variable. In 1st year 295 participants (88%) had missing data on at least one variable. A large proportion of this missing data was due to accelerometer non-compliance. Excluding MVPA, 68% of participants in 6th class had complete data, while 34% of participants had complete data in 1st year. Little’s “missing completely at random” (MCAR) test was conducted to determine the nature of missingness for data at each timepoint. Little’s MCAR test was insignificant for data in 6th class (Little’s MCAR χ^2^ (33) = 22.05, *p* = 0.93) and 1st year (Little’s MCAR χ^2^ (37) = 52.28, *p* = 0.05) indicating that missing data was MCAR [59]. Given the random nature of the missing data, a path analysis using full information maximum likelihood estimation in AMOS version 23 (IBM Corp. Armonk, NY, USA) was chosen as a means to estimate cross-lagged and autoregressive pathways between variables in 6th class and 1st year [60]. Maximum likelihood estimation is recommended for longitudinal data with missing values that are MCAR as it uses all available data for each participant to estimate model pathways [61,62].

Confirmatory factor analyses (CFA) were conducted in AMOS version 23 to create latent variables for locomotor MC, object-control MC, PAC, and HRF. First, the measurement models for each latent variable in 6th class were tested. All items tested for each variable were included in the initial CFA’s. Following statistical and theoretical analyses, catch was removed from the object-control MC latent variable due to both its relatively low loading (0.43; [63]), and the nature of its assessment within the TGMD. The success criteria for the catch include using two hands to catch a tennis ball. For older children and adolescents, advanced execution of the skill generally results in catching with one hand. Therefore, if evaluating the catch as directed in the TGMD manual, participants with more advanced skill levels who catch the ball with one hand will ultimately fail on the “catch with two hands” criteria. This indicates a clear issue with the inclusion of catch within the object-control MC latent variable when evaluating this skill in older children and adolescents. The removal of the catch resulted in an improvement in model fit in the object-control MC latent variable. All individual items were retained for locomotor MC, PAC, and HRF. Each of the latent variables showed acceptable model fit in 6th class (Table 2). Following this, measurement invariance from 6th class to 1st year was assessed and showed all latent variables to be invariant over time (Table 3).

### 2.4. Analysis

Mean values for each variable at each time point were calculated and independent samples *t*-tests were conducted to analyse sex differences for each variable in 6th class, and in 1st year. Effect sizes were calculated to determine the magnitude of any differences.

A cross-lagged regression model was specified consisting of four latent (locomotor MC, object-control MC, AC, and HRF) and one observed (MVPA) variable, each measured in 6th class and again in 1st year. Full panel analyses in AMOS, using maximum likelihood estimation, was used to test all hypotheses simultaneously. The model included autoregressive paths for each variable over time and cross-lagged paths from each variable in 6th class to each other variable in 1st year. As measurement invariance was satisfied the full cross-lagged regression model was specified with observed variables to allow for a more parsimonious model fit [64]. Variables that had a correlation of ≥0.30 within timepoints were allowed to correlate. Goodness of fit for the full model, was examined using a variety of measures including the Chi-square (χ^2^) test, the Comparative Fit Index (CFI), the Normed Fit Index (NFI), and the Root Mean Square Error of Approximation (RMSEA). A statistically insignificant χ^2^ indicates a good fitting model, although a statistically significant χ^2^ need not necessarily signify poor model fit, given that large samples have been shown to give erroneous χ^2^ results [65]. CFI and NFI values of >0.9 show acceptable model fit [66,67], while values of >0.95 for CFI are considered to show superior fit [68]. RMSEA and CFI are two of the most frequently reported fit statistics [59]. In models with few degrees of freedom (df) interpretation of RMSEA can, however, result in rejecting a good-fitting model [68].

Using the multigroup analysis, a Chi-square (χ^2^) difference test was conducted on the full model to identify any sex differences in pathways. A significant χ^2^ test indicates variance between sexes for model fit [67]. The table of critical ratios was also inspected. Values >1.96 for a given pathway indicate a significant difference between groups for this pathway [60].

## 3. Results

The mean values and sex differences for each variable at both timepoints are presented in Table 4. In both sixth class and first year, males were significantly more active, had significantly higher HRF levels, and were significantly better in object-control MC (Table 4). Males also had significantly higher PAC than females in first year, but not in sixth class (Table 4). Similarly, males were significantly better in locomotor MC in first year, but not in sixth class (Table 4). The magnitude of the differences between males and females can be interpreted from the effect size (*d)* (Table 4). According to Cohen (1992) [69] effect sizes of 0.20–0.49 are small, 0.50–0.79 are medium, and ≥0.80 are large.

Analysis of the full cross-lagged model showed a reasonable fit (χ^2^ = 69.12, df = 8, *p* < 0.01, NFI = 0.93, CFI = 0.94, RMSEA = 0.15 (CI: 0.12–0.19)). Pathways with standardised regression coefficients greater than 0.10 are shown in Figure 1.

Each variable in sixth class significantly and positively predicted itself in first year (Figure 1). HRF and object-control MC were highly stable over time, while all other variables showed moderate stability over time. MVPA was the least stable over time.

HRF was a moderate and significant predictor of MVPA (β = 0.353) and PAC (β = 0.336) in first year, and significantly predicted first year locomotor MC (β = 0.163), though this association was small. Other than the associations between sixth class HRF and first year MVPA and PAC, all other pathways from sixth class to first year variables were small (0.10–0.29). Reciprocal relationships between object-control MC and MVPA, and object-control MC and PAC were found. In addition, object-control MC in sixth class was a significant predictor of HRF (β = 0.125) and locomotor MC (β = 0.165) in first year. Locomotor MC in sixth class negatively predicted MVPA in first year, though this association was small and insignificant (β = −0.175). PAC in sixth class was a predictor of locomotor MC in first year (β = 0.100).

For variance in first-year variables, the overall model predicted 81% variance in HRF, 61% variance in PAC, 44% variance in object-control MC, 39% variance in locomotor MC, and 34% variance in MVPA. Excluding autoregressive pathways, HRF was the only 6th class variable to exhibit moderate associations with multiple variables in 1st year. Object-control MC also contributed to a variance in multiple factors in first year, but these associations were small.

Significant differences in model fit for males and females were found following a χ^2^ difference test (χ^2^ = 48.02, df = 25, *p* < 0.004). Inspection of the critical ratios table showed differences in regression coefficients for several pathways in the model (Table 5).

A moderate autoregressive pathway was found for MVPA in males, but the same pathway in females was extremely weak. A strong autoregressive pathway was found for object-control MC in females, but the same pathway for males was small. A small positive pathway from sixth class MVPA to first year object-control MC was found for males, but for females the pathway was negative, and much stronger. Similarly, the pathway from sixth class MVPA to first year locomotor MC in females was moderate and negative, whereas for males there was no visible pathway between 6th class MVPA and first year locomotor MC (Figure 2).

## 4. Discussion

The purpose of this study was to identify pathways between PA, MC, HRF and PAC, across the transition from primary to second-level school in a cohort of adolescent Irish youth. Building on previous research [8] which identified baseline HRF as a strong mediator between MC and PA, the current research sought to better understand the predictive role of each of the variables identified in Stodden et al.’s (2008) [13] model. A significant finding from this research is the role of HRF in predicting future PA, PAC and MC. Even when accounting for baseline PA, HRF was the strongest predictor of 1st year MVPA. HRF also positively predicted 1st year PAC and locomotor MC. Britton et al. (2019) [8] demonstrated that pathways were stronger in the direction of HRF to MC and PA rather than in the reverse direction. Results from the current study support these findings providing evidence for the important role of HRF in predicting future PA. Indeed, contrary to Stodden et al.’s (2008) [13] model, where MC is hypothesised as the driving force for engaging in PA, in this sample, HRF is the primary factor positively predicting MVPA post school-transition.

A positive association between HRF and PA in youth has frequently been reported, both cross-sectionally [1,70,71] and longitudinally [33,72]. Jaakkola et al. (2016) [72] found both MC and fitness in early adolescence to have a significant impact on self-report PA in young adulthood, with the association with moderate PA stronger for fitness than for MC. In the current sample, compared to MC, HRF acted as a much stronger predictor of future PA. While MC is undoubtedly an important factor for PA engagement, it is possible that HRF is more important in promoting involvement in PA. Involvement in PA can of course in turn subsequently allow for development of MC [73] and further enhancements in HRF [1]. In a study examining adolescents involvements with the ball during recreational soccer matches, Ré et al. (2016) [74] found that HRF, along with agility, was the strongest predictor of involvements with the ball, while soccer specific technical skills did not significantly contribute to an individual’s involvement. At a recreational level, having higher levels of HRF may enable youth to be more engaged within a PA setting, as demonstrated by Ré et al. (2016) [74], which may then allow for subsequent development of MC, as well as promoting higher levels of MVPA.

In comparison to other cross-sectional studies examining associations between HRF and MC [75], the longitudinal nature of the present study allows for analysis of predictive pathways between variables and clearly shows that HRF is a stronger predictor of MVPA compared to MC. Considering the potential importance of HRF in enabling youth to engage in MVPA, as demonstrated from the current findings, and the numerous health benefits associated with HRF [31], it is essential that practitioners seek to promote HRF development in youth. However, in the same way that repetitively striking a ball from a tee, such as outlined in MC test manuals, would not be recommended as a method to develop MC in youth, simply mimicking HRF tests as a means of developing HRF should also not be encouraged. Novel and fun programmes that develop HRF with a view to promoting lifelong engagement in PA are more likely to be successful [76], rather than overly prescriptive methods of HRF promotion. Programmes that take this more holistic approach seem to be more likely to succeed in their goals of increasing HRF, PA and MC [77].

In addition to HRF, object-control MC was the only other variable in sixth class that significantly and positively predicted MVPA in first year. Similar to the current results, Barnett et al. (2009) [17] found that object-control, but not locomotor MC, at age 10 significantly predicted self-report PA six years later. Object-control MC appears to be more stable over time compared to locomotor MC [24]. The relationship between object-control MC and MVPA in the current sample was reciprocal, supporting the hypothesis outlined in Stodden et al.’s (2008) [13] conceptual model. This is in keeping with a previous study which reported a positive reciprocal relationship in older adolescents for object-control MC and self-report MVPA [78]. Consistent with Barnett et al.’s (2011) findings, the current study found no reciprocal relationship between MVPA and locomotor MC. In the current sample object-control MC, more so than locomotor MC, was predictive of future PA. Object-control MC encompasses skills such as kicking, throwing, and striking. These skills are required for participation in many of Ireland’s most popular youth sports [79]. Despite other forms of activity available to young people which require locomotor proficiency (cycling, walking, running), it is likely that activities which require object-control MC are more popular in an Irish adolescent population, and that, without proficiency in this component of MC, youth are likely to disengage from popular sports and thus fail to be sufficiently physically active for health. In addition to a reciprocal relationship between object-control MC and MVPA, a reciprocal relationship was also found between object-control MC and PAC. While PAC did not predict MVPA in this sample, similar to other studies using objective measures of PA [40], it did positively predict object-control MC. In fact, this reciprocal relationship was slightly stronger in the direction of PAC predicting object-control MC rather than the reverse. It may be that higher levels of PAC provide young people with the confidence to develop their skills, and that, in time the knock-on effect of higher PAC on MC would then promote future engagement in MVPA. Further research is required to fully investigate this pathway in adolescence.

### Differences Between Males and Females

Substantial differences in some of the pathways depicted in the model were seen between males and females in the current study. For males, MVPA in sixth class was a strong predictor of first year MVPA, whereas for females this pathway was virtually non-existent. This supports previous research nationally [80] and internationally [81] which has found PA to be more stable in males compared to females over time. In the current study, PA was less stable across the school transition compared to HRF, MC and PAC. PA is a behaviour, whereas HRF and MC are individual attributes [82]. Compared to HRF and MC, PA is more likely to be influenced by psychosocial factors [83], which can be magnified during major transitions such as the school transition [81]. It is possible that the instability in PA seen in females during the school transition reflects a greater impact of school transition on females compared to males. In research on the effects of the school transition on adolescents, Anderson et al. (2000) [84] found that the impact of the transition on self-perceptions was greater on females. It may be that the same holds true for PA behaviours during this transitional period.

While MVPA was less stable in females than males across the transition, the opposite was found for object-control MC, indicating that object-control MC is less likely to change in females compared to males across the school transition. A longitudinal study in Irish youth reported that, between the ages of 10 and 18, females were more likely to drop out of organised sport and less likely to take up a new sport compared to males [79]. Popular youth sports in Ireland, such as Gaelic games, soccer, and basketball [79] provide the opportunity to develop many object-control skills. If females are more likely to drop out of sport during the adolescent years [79], then it is likely they will not have the opportunity to further develop their object-control MC, potentially explaining the higher stability in this MC component over time.

Contrary to expectation, sixth class MVPA negatively predicted first year MC for females. In comparison, sixth class MVPA did not have a substantial effect on first year MC for males. MC has been identified as an important factor for social acceptance within peer groups for males [85]. For females however, no positive association is seen between MC and popularity with peers [85]. In fact, earlier studies found that sport competency was associated with being the least popular member in female peer groups [86,87]. This could potentially explain how MVPA in sixth class negatively predicted MC in first year for females in this sample. Female participants who were relatively active in primary school may recognise that being competent in sports or motor skills is not in fact conducive to being accepted within their peer group, and as a result may depress their abilities simply to “fit in”.

That being said, HRF in sixth class was a positive predictor of MC for females in first year. HRF in females, and males, is highly stable over time. In comparison, MVPA for females is highly unstable. This lack of stability in MVPA compared to HRF for females may in part explain how MVPA could negatively predict MC while HRF positively predicts MC. Notwithstanding the significant mean change in MVPA in females across the school transition, significant intra-individual change is also occurring, as indicated by the low stability of MVPA. Thus, females who were highly active in sixth class may be low active in first year and vice versa. This may then have a knock-on effect on their execution of MC in first year. In comparison, HRF in females is much more stable indicating that fitness levels attained by sixth class of primary school carry through into first year of second-level school. In addition, compared to MVPA which is a singular variable, HRF is a multifaceted construct [26,88]. Having high levels of HRF indicates that an individual displays fitness across a range of different activity-types (e.g., activities requiring CRE, MS and ME). Higher levels of HRF in sixth class likely provide individuals with the tools to engage in numerous activities in first year that require proficiency in various FMS, thus enabling the development of MC [82]. It is unlikely that females who were active in 6th class lose the capacity to successfully execute the basic components of many object-control and locomotor skills in first year. It is more likely that the perceived value of MC and PA within the female peer group impacts on the desire of some female adolescents to appear competent in what is often perceived as a male environment [85].

## 5. Conclusions

This is the first study to explore the predictive pathways between all of the variables depicted in Stodden et al.’s (2008) [13] model across the school transition. In addition, the use of CFA to check for measurement invariance in each variable over time in this study is a crucial yet often ignored practice in longitudinal analyses. Poor accelerometer compliance, and the subsequent loss of PA data, along with a high degree of missing data in general, is a limitation of this study, not uncommon in longitudinal research in this field. However, the use of full information maximum likelihood path analysis allowed for analysis on pathways between variables from primary to second-level school to be conducted, despite missing data.

The findings from the current study point to the importance of HRF for future PA engagement in adolescents. Whereas MC is commonly positioned as one of the primary predictors of PA, this study found HRF, measured as a composite, to be more important in predicting PA across the transition from primary to second-level school. Object-control proficiency, as a component of MC, was however found to have a positive effect on PA, and other factors shown to be associated with positive PA behaviour. From this perspective it would seem that developing HRF and object-control MC in primary school may be crucial factors to consider as we strive to address the frequently reported decrease in PA observed across the school transition.

## Figures and Tables

**Figure 1 ijerph-17-02874-f001:**
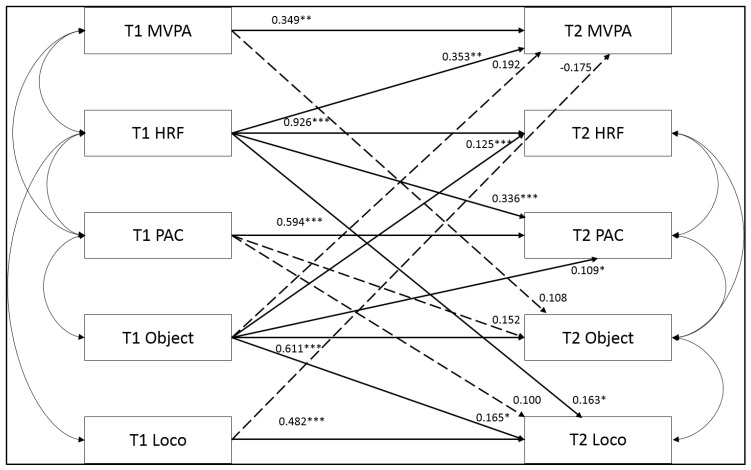
Autoregressive and cross-lagged pathways between variables in 6th class (T1) and 1st year (T2). Note: * Standardised regression coefficients significant at *p* < 0.05; ** *p* < 0.01; *** *p* < 0.001. Dashed line = non-significant pathway.

**Figure 2 ijerph-17-02874-f002:**
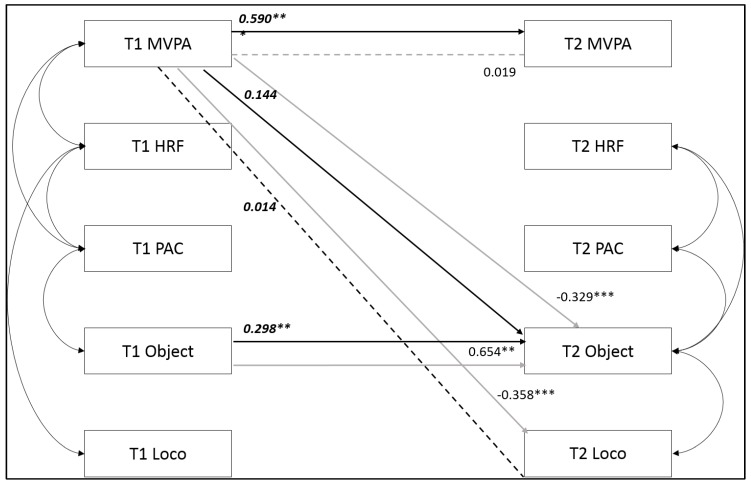
Pathways between variables from 6th class (T1) to 1st year (T2) that were statistically significantly different between males and females. Only significantly different pathways are shown. All other pathways were not statistically different by sex. Note: * Standardised regression coefficients significant at *p* < 0.05; ** *p* < 0.01; *** *p* < 0.001. Grey line = female pathway; black line = male pathway. Dashed line = non-significant pathway. Regression coefficients shown in *italics* = male values.

**Table 1 ijerph-17-02874-t001:** Measurement of health-related fitness (HRF).

HRF Component	Test	Source
CRE	20 MST	FITNESSGRAM; EUROFIT
MS	HJ	EUROFIT; HELENA Study
**-**	VJ *	HELENA Study
ME	Push-ups	FITNESSGRAM
-	Curl-ups	FITNESSGRAM

Note: CRE = cardiorespiratory endurance; MS = muscular strength; ME = muscular endurance; 20 MST = 20 metre shuttle run test; HJ = horizontal jump; VJ = vertical jump; FITNESSGRAM [55]. The HELENA Study [27]. * VJ was assessed using the Abalakov jump test protocol outlined in the HELENA study [27,57] and using a jump mat and belt [58] in place of an infrared jump platform.

**Table 2 ijerph-17-02874-t002:** Fit indices for latent variables in 6th class.

Variable	χ^2^	Df	*p*-Value	CFI	RMSEA	LCI	UCI
PAC	5.56	5	0.352	0.990	0.040	0.020	0.070
Object-Control *	1.1	2	0.576	1.000	0.000	0.000	0.090
Locomotor	2.65	2	0.266	0.980	0.031	0.000	0.117
HRF	14.26	3	0.003	0.960	0.106	0.055	0.164

Note: * catch removed due to low loading. χ^2^ = Chi-square; Df = degrees of freedom; CFI = Comparative Fit Index; RMSEA = Root Mean Square Error of Approximation; LCI = lower confidence interval; UCI = upper confidence interval; PAC = perceived athletic competence; HRF = health-related fitness.

**Table 3 ijerph-17-02874-t003:** Measurement invariance for latent variables.

Variable	χ^2^	Df	*p*-Value
PAC	6.33	4	0.176
Object-Control *	7.33	3	0.062
Locomotor	5.291	3	0.152
HRF	7.786	4	0.100

Note: * catch removed due to low loading; χ^2^ = Chi-square; Df = degrees of freedom; PAC = perceived athletic competence; HRF = health-related fitness.

**Table 4 ijerph-17-02874-t004:** Descriptive statistics for measured variables.

	Male		Female		*t*	*p ^a^*	*d ^b^*
Testing Period	*n*	Mean	*n*	Mean			
6th class MVPA (mins)	58	59.27 ± 24.55	77	48.50 ± 16.82	2.87	0.005	0.52
1st year MVPA (mins)	44	44.95 ± 18.83	59	35.11 ± 12.12	3.04	0.003	0.64
6th class HRF	128	1.32 ± 3.71	133	−1.27 ± 3.08	6.12	<0.001	0.76
1st year HRF	155	1.41 ± 3.43	143	−1.51 ± 2.65	8.22	<0.001	0.95
6th class PAC	122	2.92 ± 0.67	125	2.75 ± 0.73	1.94	0.053	0.24
1st year PAC	135	2.92 ± 0.70	119	2.58 ± 0.79	3.6	<0.001	0.46
6th class object-control MC	120	27.91 ± 4.01	118	23.12 ± 5.55	7.62	<0.001	0.99
1st year object-control MC	77	28.88 ± 3.51	62	24.90 ± 5.08	6.57	<0.001	0.92
6th class locomotor MC	119	29.29 ± 3.85	119	29.02 ± 3.64	0.55	0.580	0.07
1st year locomotor MC	99	30.78 ± 3.21	86	24.90 ± 5.08	2.46	0.015	1.40

**Note:**^a^*p*-value for sex differences; ^b^ Cohen’s *d* effect size; MVPA = moderate-vigorous physical activity; HRF = health-related fitness; PAC = perceived athletic competence; MC = motor competence.

**Table 5 ijerph-17-02874-t005:** Pathways by sex.

	Males		Females	
Pathway	β	*p*	β	*p*
T1 MVPA–T2 MVPA	0.590	***	0.019	NS
T1 Object-Control–T2 Object-Control	0.298	**	0.654	***
T1 MVPA–T2 Object-Control	0.144	NS	−0.329	**
T1 MVPA–T2 Locomotor	−0.004	NS	−0.358	**

**Note**: β = standardised regression coefficient; * significant at *p* < 0.05; ** *p* < 0.01; ****p* < 0.001; NS = non-significant; T1 = 6th class; T2 = 1st year, MVPA = moderate-vigorous physical activity.

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
