# Peer review of "What Keeps Them Physically Active? Predicting Physical Activity, Motor Competence, Health-Related Fitness, and Perceived Competence in Irish Adolescents after the Transition from Primary to Second-Level School"

_ijerph, 2020, doi:10.3390/ijerph17082874_

Round 1

Reviewer 1 Report

Dear authors, I would like to point out three aspects that would improve the presentation and clarity of the document:

  1. In the introduction and theoretical framework section, the hypotheses should be listed and specified. For example: H1: text of the hypothesis
  2. It is recommended to include in the results the effect size associated with t-tests (Cohen's d). Specify the inclusion of this analysis in the method section.
  3. In the path diagram of Figure 1, it is recommended to put the relationships between variables that are not significant with a dashed line

Author Response

Dear Reviewer,

Thank you for taking the time to review this article, and for the invaluable feedback you have given thus far.

Please see attached an outline of the changes which address your most recent review.

Kind regards,

Una Britton.

Reviewer 2 Report

Dear Authors,

The manuscript has been revised again and I see that it has been improved. Well Done.

Author Response

Dear Reviewer,

Thank you for taking the time to review this paper, and for the extremely helpful feedback.  

Kind regards,

Una Britton.

This manuscript is a resubmission of an earlier submission. The following is a list of the peer review reports and author responses from that submission.

Round 1

Reviewer 1 Report

First, authors should check the citations in the text because there are quite a few that do not follow the guidelines of the journal. On the other hand, they should also review the formatting aspects because they have not fully followed the journal's guidelines.
In tables 1, 2 and 3 you should put "Note" before explaining the legend. Please review the journal template for writing articles.
Figure 1 should include a note indicating the meaning of the asterisks that appear next to each numeric data.
It is unclear whether informed consent was given to the parents of the students who joined the study (n=75) in the 1st year sample. Please clarify this in the participant section.
On the Harter scale, additional measures of the reliability of the constructs under study are missing, such as composite reliability and mean variance extracted.
Why is the RMSEA value and its confidence interval not shown for the structural equation model?
What is the meaning of CMIN on line 247? When including acronyms, please explain their meaning the first time.
There is no table note in Table 4 that explains the meaning of acronyms and asterisks.
The results should be completed by indicating the descriptive results of the variables under study.
The references do not follow the journal's guidelines.

Reviewer 2 Report

  1. Abstract: (a) The study design should be clearly stated; e.g. The aim of this study was to examine longitudinal cross-lagged relationships between....etc (Line 15) (b) It is better that this section does not contain any reference(s) (c) Specify where this research has been conducted (including in the title).
  2. What’s the theoretical basis for stratifying the models by gender for the cross-lagged correlations? This ought to be explained in the introduction.
  3. Results: I would suggest including two more cross-lagged paths (for males and females). In both models, the value of associations and significant differences should be written and marked with one asterisk (*) if p<0.05 and two (**) if p<0.01….etc.
  4. The reference list needs formatting in accordance with IJERPH.